# Anticipated Memories and Adaptation from Past Flood Events in Gregório Creek Basin, Brazil

**Hailton César Pimentel Fialho** [1,*] **, Fernando Girardi Abreu** [1] **, Bruno José de Oliveira Sousa** [1] **,**
**Felipe Augusto Arguello Souza** [1] **, Namrata Bhattacharya-Mis** [2] **,**
**Eduardo Mario Mendiondo** [1] **and Paulo Tarso Sanches de Oliveira** [3]

1   Department of Hydraulics and Sanitation, São Carlos School of Engineering, University of São Paulo,
    São Carlos 13566-590, Brazil; abreu.fernando@gmail.com (F.G.A.); brunosousa@usp.br (B.J.d.O.S.);
    felipeaas@usp.br (F.A.A.S.); emm@sc.usp.br (E.M.M.)
2   Department of Geography and International Development, University of Chester, Chester CH1 4BJ, UK;
    n.bhattacharyamis@chester.ac.uk
3   Faculty of Engineering and Geography, Federal University of Mato Grosso do Sul,
    Campo Grande 79070-900, Brazil; paulot@sc.usp.br
*   Correspondence: hailtoncesar.pa@usp.br

**Abstract:** In this research we used walking interviews to investigate the measures used by shopkeepers as protection against floods. The concept of anticipated memory has been used to identify the relationship between their learning from previous events and the adaptive measures they have taken to reduce risk of future flooding in Gregório Creek basin. The area is affected by major flooding issues in the city of São Carlos, southeastern Brazil. Twenty-three (23) downtown merchants shared their experience of the extreme rainfall that occurred on 12 January 2020, characterized by a return period of 103 years. Comparing our findings with November 2015 and March 2018 floods (Interviews 37 and 52 respectively), we noted that due to the enhanced level of threat, people had changed their adaptation strategy by increasing the sum of floodgate height more than 4-fold (870 cm to 3830 cm) between 2015 to 2020. Our results showed that despite frequent flooding, the shopkeepers downtown were reluctant to move away from the area; rather, they preferred to improve their individual protection. The substantial increase in the height of the floodgates represents the population's feedback in the face of a new level of threat.

**Keywords:** flood risk; socio-hydrology; memory; flood protection; shop keepers

## 1. Introduction

Floods are the type of disaster that causes the most material and human losses worldwide. Despite billions of dollars of investment in disaster risk reduction, the damage has been aggravated by uncontrolled urbanization and evident climate change [1]. In developing countries, the effects of floods are exacerbated by unplanned urban growth, and a lack of public policies and appropriate risk management measures [2]. Due to the impossibility of completely eliminating floods, many countries seek to devise ways to deal with losses [3]. In Brazil, according to the National Center for Monitoring and Alerts for Natural Disasters-CEMADEN, there are currently 40,000 mapped risk areas in the 958 monitored municipalities, where 3986 correspond to flood risk areas [4].

Rapid recovery is extremely important, because delays may increase social and economic pain [5]. Di Baldassare [6] claims that, in flood risk assessment, there is an adaptive effect related to the observation that the higher frequency of floods is also associated with the decrease in vulnerability. The author found empirically that the impacts of a flood event are less when events of the same magnitude occur after short periods of time. For the author, this effect can be attributed to the ability to adapt acquired during such flood events. Thus, it is considered that future projections related to flood risk may provide unrealistic information about the situation.

Risk perception is formed by sources of information, changed by individual experience, and modified by socio-economic and cultural environment. It is a way of expressing uncertainty and accumulating individual perception about natural hazards [7]. Recent study has shown that there are four expected types of society that deal with floods [8]:

i.      risk neglecting societies: such societies do not adopt protective measures because they believe it is impossible to manage risk;

ii.     risk monitoring societies: they use inclusive bottom-up strategies to address their risk;

iii.    risk downplaying societies: they underestimate environmental risk and do not see the need for collective actions; and

iv.     risk controlling societies: such societies rely on protective measures such as increasing the height of barriers after extreme events.

Furthermore, it has been said that societies that rely on top-down hierarchies and structural measures as protection against flood may still suffer high losses in extreme events. Meanwhile, societies that invest in increasing awareness through inclusive and participatory approaches experience reduced flood losses [8]. According to Garde-Hansen et al. [9], for communities to be aware of flooding (and prepare and act) as a form of socio-ecological resilience, they need to record memories and remember events in a proactive manner. According to [10], memory is described as the accumulated experience based on the history of a system, and provides the basis for identifying sources of renewal, innovation, recombination, self-organization and innovation after stressful conditions.

Environmental psychologists explain that there are two effects related to flood: the crisis effect and the levee effect. The first indicates that disaster awareness peaks immediately after flood occurrence, and dissipates between disasters. The second refers to the fact that once protection measures have been taken, people seem to be lulled into thinking that their protection will protect them against all future floods [11].

Righetto et al. [12] carried out a survey of the damage after a flood recorded in the municipal market region (the same considered in the present study), located in the Gregório Creek basin, in January 2004 (São Carlos—SP), through interviews with 14 shopkeepers to understand their level of resilience. Although the frequency of flood events has increased during recent decades in Brazil, only a few measures have been considered to address this hazard, and very few are effective.

The objective of this study is to investigate whether the measures used by shopkeepers as protection against floods are related to acquired memories from previous events in a basin with major flood problems in São Carlos, southeastern Brazil. Urbanization started in the municipality in the 18th century in the western part of the Gregório Creek catchment. Mendes and Mendiondo [13] divided the period after the first flood record into three stages, related to the urbanization rate in the catchment. In the first stage is from 1950 to 1970, the second from 1970 to 1980, and the third from 1980 to 2002. Sarmento [14] introduced a fourth stage ranging from 2004 to 2018, during which the urbanization rate nearly doubled. From the first stage until the fourth, the urbanized area increased from three to fourteen square kilometers, approximately 75% of the total area. The studied basin suffers from constant flood events, and the area is the most important commercial center of the city, with no presence of residential properties/homeowners. The area also suffers from frequent flooding, and the shopkeepers in the area have had previous experience of flooding. We address the population's preparedness against these events and also show the measures shopkeepers have taken as protection tools. Three flood events are used to build a scenario, 2015, 2018 and 2020, with return periods of 17, 8 and 103 years, respectively. Our study shows how people can respond to a new level of threat from flooding and how the accumulation of memories from previous floods improves individual protection.

## 2. Materials and Methods

The Gregório Creek basin is located in the city of Sao Carlos—SP, in Brazil's southeast region, and has a total area of 18.9 km$^2$ [15]. Floods have a greater impact in the commercial area of the municipality of São Carlos (see Figure 1). According to the Köppen climate

classification system, the climate in the studied watershed is Cwa: humid temperate, with a dry winter from June to September and a hot and rainy summer from December to March; the predominant soil class is purple, dark red and yellow red latosol [16]. It has an average annual total rain of 1361.6 mm, with the month of January being the rainiest, with 274.7 mm, on average, and August the least rainy, with 22.8 mm, on average [17].

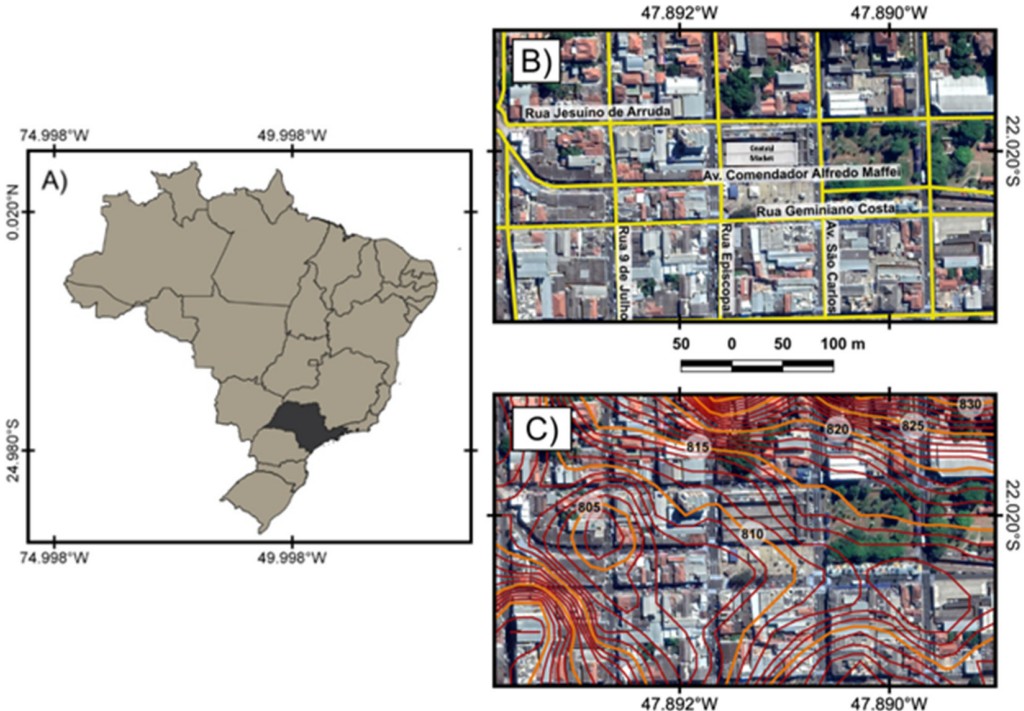

**Figure 1.** Map of the case study. (**A**) Location of São Carlos city. (**B**,**C**) Region affected by the floods. (**B**) Street names surrounding the area; (**C**) topography of the region.

In this research, we consider memory to be a key component to improving the resilience of a system; according to Wilkinson [18], resilience is the capacity of a system to renew, reorganize and regenerate from stress. It can encapsulate the process of response to challenges or adversity, positive outcomes of coping, and the capacity to create adaptive responses to obtain desired outcomes [19]. To collect data for resilience, the use of interviews, questionnaires and surveys emerges as an important methodology. The use of surveys to acquire information about social responses to extreme events is not new; McBean et al. [20] applied a questionnaire to seven communities in the province of Ontario, Canada, which included registration information of the people affected, as well as the type of structure, description of the content of each part of the residences, and the costs of goods, for a total of 287 interviews. Smith and Freedy [21] used a questionnaire to examine the role of psychosocial resource loss in the aftermath of floods among 131 participants who lived in eight flooded communities along the Missouri and Mississippi rivers. Terpstra and Gutteling [22] applied an Internet survey in the province of Friesland in The Netherlands, to explore Dutch households' perceived responsibility for taking private protection measures against floods. A questionnaire survey of 719 randomly sampled respondents was conducted in 16 sub districts of Jingdezhen to investigate the public flood risk perception by Wang [23]. In Brazil, Lima [24] constructed histograms on the basis of interviews with the affected population in the city of Itajubá, Brazil, presenting the results in terms of average values of flood damage as a function of submerged depth ranges.

In this research, a survey was created on the Google Forms platform, which was of great help, since it is capable of generating reports from the collected data, providing predetermined automatic calculations, the possibility of including photographs of the location, as well as the collection of geographic coordinates and easy export to Excel.

The questionnaire survey was conducted with the owners or managers of commercial properties, and the research was built upon the previous findings of Abreu's surveys in the same area [25]. Abreu [25] performed interviews in 52 stores (Figure 2) from April to May 2019, addressing the extreme events of 23 November 2015 and 20 March 2018, and among which only 37 stores were located in the same place as in 2015. The Central Market wasn't included in the questionnaire, because they did not suffer great losses due to having a gate at the entrance of the building to protect it from floods, while the stores inside the market have their own floodgates. The major damage happened to the stores on the streets.

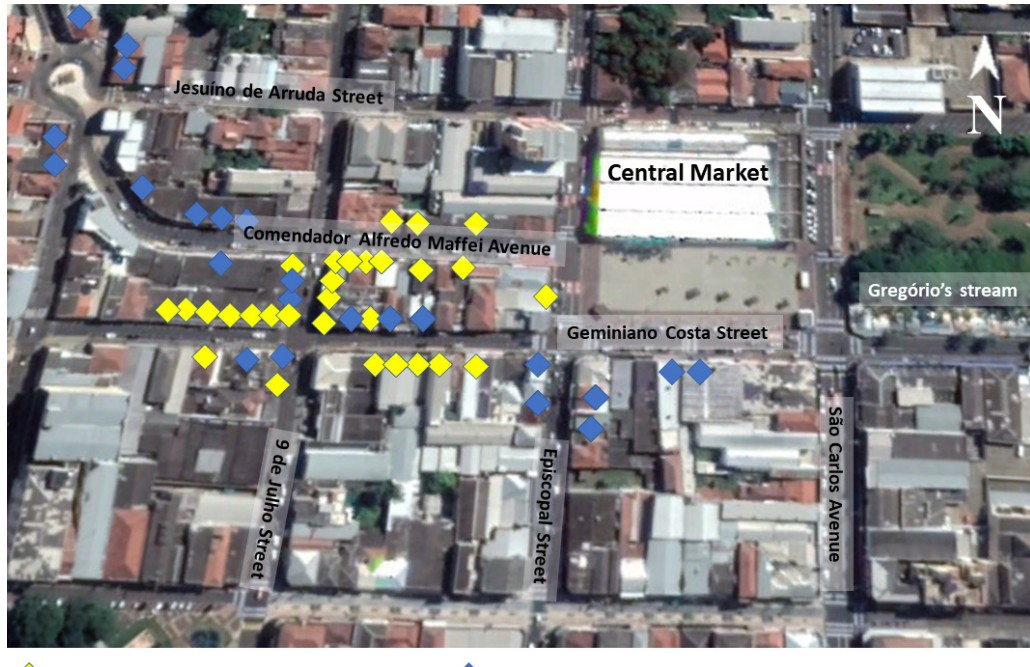

**Figure 2.** The 52 stores interviewed by Abreu [25].

In January 2020, we conducted walking interviews in 23 stores about the flood event of 12 January 2020. The 52 stores chosen by Abreu [25] and the 23 in this work were chosen from among 141 stores potentially suffering from damage with the extension of the flooded part of the floodplain (based on observational data). The questions used are shown in Table 1. The water level height inside and outside the store were measured using the store's floor and the sidewalk in front of the store, respectively, as a reference. To classify the basin as risk neglecting, risk monitoring, risk downplaying or risk controlling, we performed a cluster analysis using the K means method, which is a clustering algorithm that groups data on the basis of a cluster center point closest to the data. Its purpose is to group data with maximum similarity into one cluster and minimize the data similarity between clusters [26]. The questions used in the cluster analysis were "HOW LONG HAS THE ESTABLISHMENT EXISTED AT THIS ADDRESS? (YEARS OLD)" and "FLOODGATE HEIGHT (cm)".

**Table 1.** Questionnaire related to flood impacts in the commercial center of São Carlos/SP.

| |
|---|
| 1. Name |
| 2. Address |
| 3. Commercial Activity |
| a. Department stores, warehouse items, clothing and fabrics, shoe and leather prducts<br>b. Furniture, electric household appliances, computer equipment<br>c. Pharmaceutical products, perfumeries and cosmetics<br>d. Jewelry shops, watches and optics<br>e. Restaurants, anacks, bars and coffee<br>f. Newsstand and stationery |
| 4. Ground Plan Height in Relation to the Sidewalk (cm) |
| 5. Water Level Inside the Establishment (cm) |
| 6. Water Level Outside the Establishment (cm) |
| 7. How Long does the Establishment Exist at this Address? (Years old) |
| 8. Does It Have a Floodgate?<br>a. Yes<br>b. No |
| 9. Floodgate Height (cm) |
| 10. Willingness to Pay for a Flood Insurance a Year (U$) |
| 11. General Observations |

The choice of the 3 events, 2015, 2018 and 2020, was made due to their magnitude, which caused a lot of inconvenience and damage to shopkeepers in the region (see Figure 1), which is a densely urbanized area, with several commercial establishments located on the banks of the Gregório Creek, that deserves greater attention. The decision to include the flood of 2015 was crucial to this study, because it marks a change in behavior, with the adhesion and increasing of floodgates as a protective measure against flood. The analysis of historical series was performed using the pluviograph 354890601A [27], located in the Gregório Creek basin, and the return period was determined on the basis of the most used local IDF equation for the municipality of São Carlos [28].

In addition, another field survey was carried out in April 2020 to collect floodgate height data and determine which stores had closed or changed addresses among the 52 stores interviewed by Abreu [25] in 2019. This field survey was important to show how stores stores had reacted after 2020's flood and to check whether there had been any improvement in their individual levels of protection, since the interviews did not capture any post-2020 adaptation, due to the short time between the flood and interview. Figure 3 shows a timeline with the flood events, and the conducting of the interviews and field survey. Regarding the geographic organization of the area, Rua Episcopal (Episcopal street) is the 1st floodplain transect to the perpendicular creek. Following is the Rua 9 de Julho (9 de Julho street), which is the second floodplain transect perpendicular to the creek. Av. Comendador Alfredo Maffei (Comendador Alfredo Maffei Avenue), on the other hand, runs above the river from the intersection with Av. São Carlos (São Carlos avenue). Finally, Rua Geminiano Costa (Geminiano Costa street) is the first street parallel to the creek at the left margin (Figure 1B).

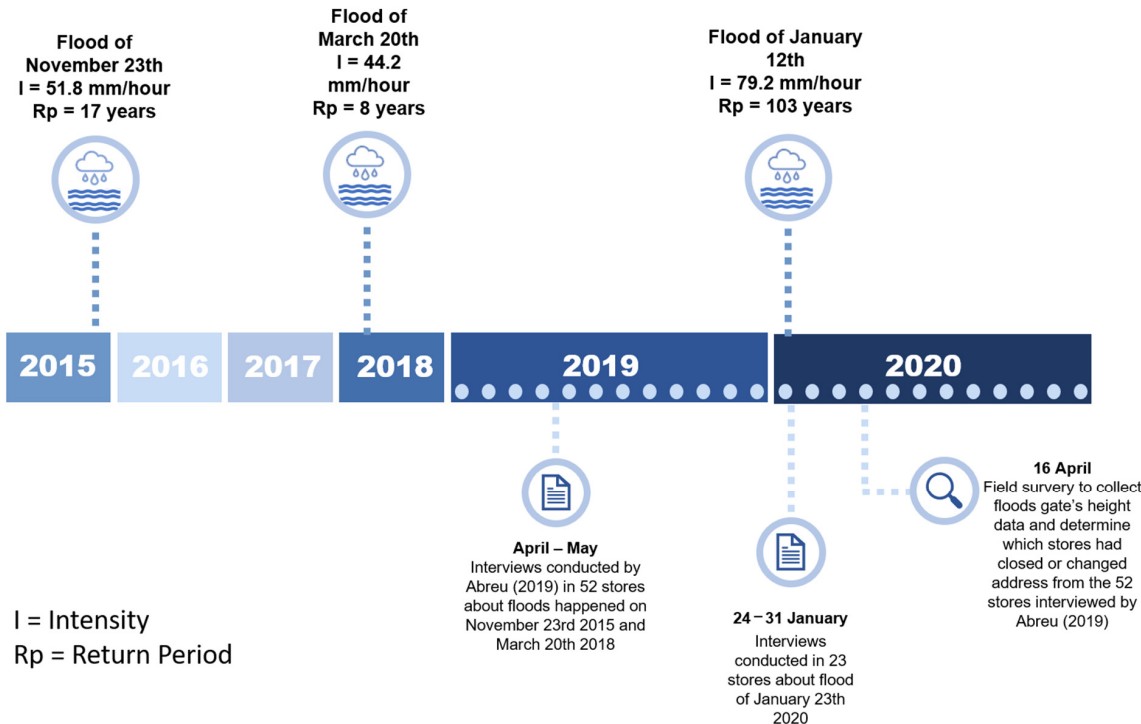

**Figure 3.** Timeline of floods considered, and the interviews and field survey conducted.

## 3. Results and Discussion

### 3.1. Ideal Types of Society

The Gregório Creek basin has faced several flood events since urbanization started, and few measures have been implemented to increase people's awareness of and preparedness for extreme events [14]. Real-world societies consist of mixtures of the four ideal types of societies described by Ridolfi [8]. Figure 4 shows the distribution of the interviewees into four clusters, with 1 and 2 having the most representativeness, followed by 3 and 4, respectively.

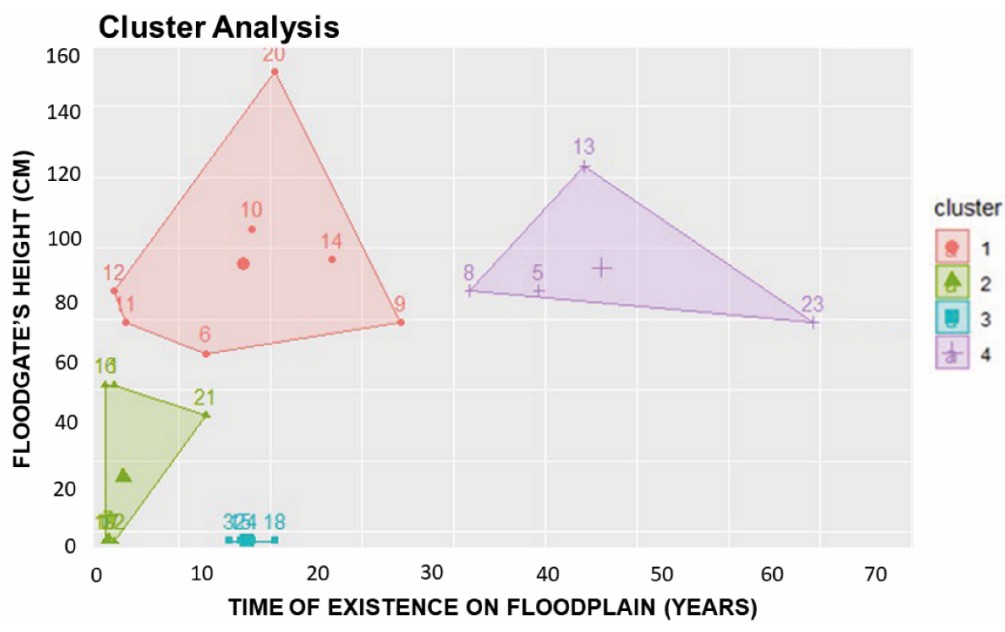

**Figure 4.** Cluster analysis of ideal types of society.

Cluster 1 has seven stores (30%), with floodgate height and existing time varying from 60 to 150 cm and 1 to 26 years, respectively. It can be classified as a risk controlling society, characterized by a top-down perspective and the use of structures as protection measures that increased in height over time. Events with water levels lower than the protection were quickly forgotten and no major losses were recorded, while extreme events caused high losses and refreshed their memory, which decreases for as long as no flood is experienced, creating a feeling of safety [6,8]. Cluster 2 has seven stores (30%), with floodgate height and existing time varying from 0 to 50 cm and 0.2 to 9 years, respectively. This cluster can be classified as a risk downplaying society that underestimates hydrological extremes, and having no intention of relocating in the case of flood occurrence. Cluster 3 has five stores (22%), did not have floodgates, and existing time varied from 11 to 15 years. It can be classified as a risk neglecting society, characterized by not adopting any measure of preparedness. The population indicated that they had no intention of moving to another location, regardless of whether the settlement was on a floodplain. The main reason for this was the fact that the study area has a very strong commercial identity; therefore, changing locations could lead to a reduction in the number of customers. Cluster 4 has four stores (17%), with floodgate height and existing time varying from 70 to 120 cm and 32 to 62 years, respectively. Its classification is as a risk monitoring society. This society is aware of being in a precarious balance that can be disturbed if risk is not addressed correctly. It knows that its needs have to change, and not nature, which had caused two stores to relocate to places with less risk.

### 3.2. Characteristics of the Flood and Its Consequences

This research shows how the population dealt with the extreme events that occurred in 2015, 2018 and 2020, and the measures used after each event. For the 2015 event, the rain intensity in the most intense hour was 51.8 mm/hour, which corresponds to a return period of 17 years. In the event of 2018, the most intense hour recorded was 44.2 mm/hour, which corresponds an 8 year return period. In 2020, the highest intensity observed was 79.2 mm/hour, which is equivalent to a return period of 103 years. Despite the floods, until 2018, all 52 merchants interviewed declared that they did not intend to leave the site, since it is an important commercial spot in the city. However, after the 2020 event, due to its magnitude, nine establishments (17%) closed and three (6%) changed addresses to a less risky area.

On the basis of the interviews in 2020, it was observed that the main factor that caused the losses was the magnitude of the event, because, according to reports, never before had there been such a large amount of rainfall in such a short time—more than 92 mm in three hours of precipitation, of which 86% occurred in the first hour. The merchants, in all surveys, stated that the main factor in the level of damage of the event was the height and speed of the water, causing the dragging of objects and vehicles that were on the street. The owner of a home appliance maintenance store said that the door had always prevented flooding inside the store, but a cart was dragged and hit it, allowing water to enter, reaching a height of 120 cm, causing damage not only to the merchandise, but to the furniture and to a vehicle that was kept inside the store. Water also invaded the establishment through the plumbing system of the building, entering through the drain and toilet. The building is rented, and the store owners are thinking of changing location.

It is worth noting that the owner of a traditional store registered a water height of 125 cm inside the store; water managed to invade the establishment even with the 120 cm gates closed. He reported that he had been in this location for 42 years, and that it was the biggest flood he had experienced. The same owner said he will change his address, because in January 2020 alone there had been three flood events—on 2 January, 4 January and 12 January—with the latter being the most intense, registering 170 cm of water in front of his store. Another owner of a shoe store also reported that in the 38 years that he had owned the establishment, the flood on 12 January 2020 was the most damaging.

The flood that occurred on 12 January 2020 was used to build a social flood map (see Figure 5). To create it, a field visit was made to delimit the flooded part of the floodplain on the basis of the data collected from traders' reports, which showed the height that the water had reached at each point and how far the flood went into the basin. This map accurately represents the flood locations, as the data collected were fresh in the memories of the traders, as the area was still under repair by the government. We talked to shopkeepers to schedule walking interviews, because several merchants did not want to talk about the flood, as they were still dealing with the damage and trying to reorganize their businesses.

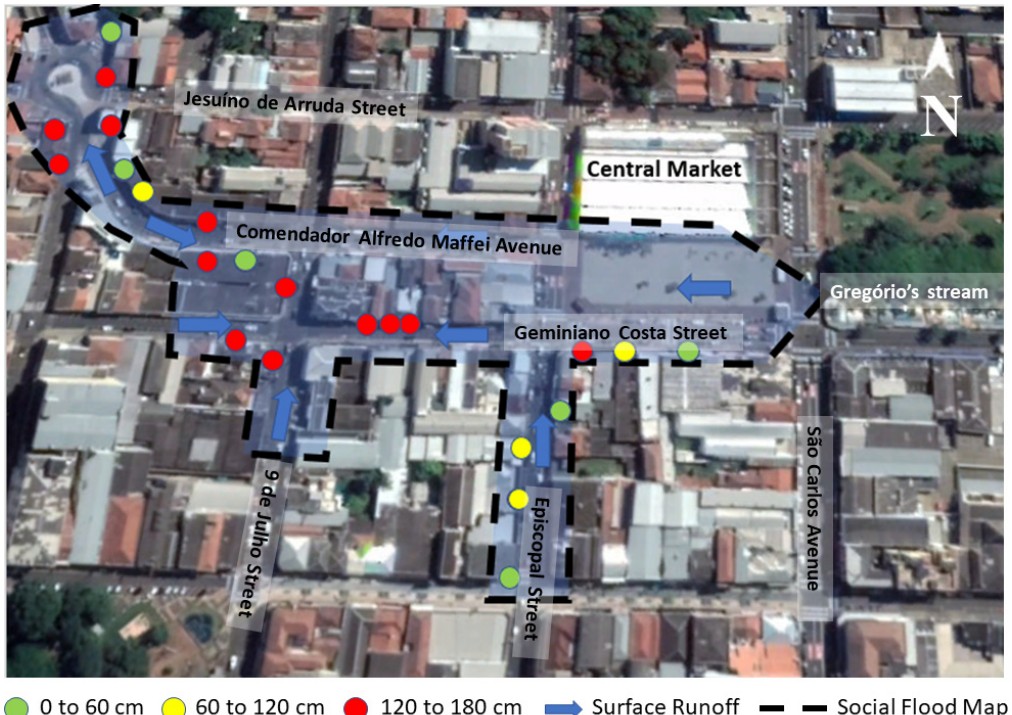

● 0 to 60 cm    ● 60 to 120 cm    ● 120 to 180 cm    ➡ Surface Runoff    ▬ ▬ Social Flood Map

**Figure 5.** Social flood Map. The colors represent an external height range reached by water according to voluntary data. The dashed line represents the flooded part of the floodplain during the 12 January 2020 flood.

### 3.3. Protective Measures and Water Level

The protective structure most used by shopkeepers was floodgates (Figure 6). According to surveys conducted by Abreu [25] and the field survey in 2020 in the flood area, from a sample of 37 stores, in 2015, 12 (32%, average 77.5 cm) had floodgates. In 2018, from a sample of 52 stores, 31 (61%, average 92.7 cm) had gates. In 2020, from the same 52 stores surveyed by Abreu (2019), 39 (75%, average 102 cm) had floodgates. Figure 7 presents a statistical analysis in the form of a box plot of the averages of the installed gates in relation to the respective flood event. The graph shows that, even if the heights measured in 2015 were within the height intervals found in 2018 and 2020, the median for that year was below the median values measured after the other two events. Therefore, an increase in floodgate height can be observed after 2015.

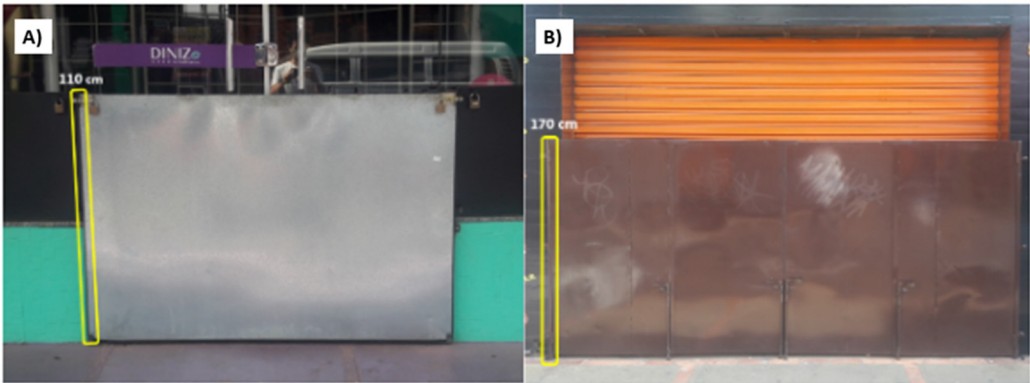

**Figure 6.** Example of floodgates installed. (**A**) A store located in Comendador Alfredo Maffei Avenue; (**B**) a store located in Geminiano Costa Street. These pictures were taken on 10 October 2020.

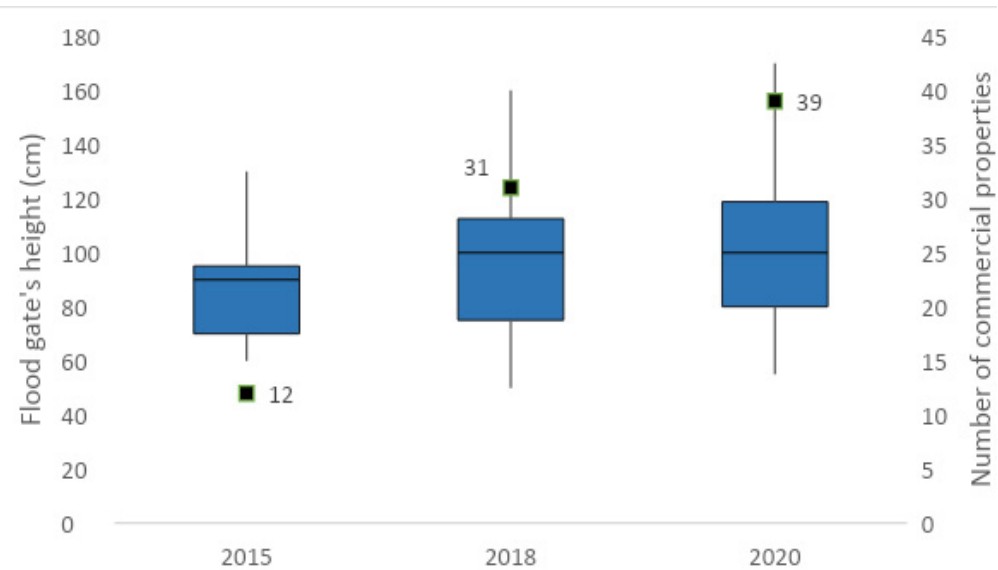

**Figure 7.** Variation in the height of the gates installed in the stores analyzed between the years 2015, 2018 and 2020. The square shows the number of stores with gates among the 52 stores studied each year.

Figure 8 presents the sum of the floodgate heights of the properties whose owners were interviewed after the events of 2015, 2018 and 2020, discretized according to the respective streets in which they are located. This representation assesses the population's preparation for a new threat in view of the experiences accumulated in previous events. We noted that the height of the gates in 2015 was smaller than in 2018, and both were smaller than in 2020, with the total sum of the height of the gates in 2015, 2018 and 2020 being 870 cm, 2815 cm and 3830 cm, respectively. It was noted that, as a mitigation measure after 2015, of the 37 stores, 16 (43%) installed floodgates and 3 (8%) increased the level of their existing floodgates. After 2018, of 52 stores, 9 (17%) installed floodgates, and 7 (13%) increased the level of their existing floodgates. This substantial increase in the cumulative height of gates (Figure 7) and the average height of these gates represents the population's response to a new threat level. This means that when they face an extreme event, they respond by increasing the height of floodgates in order to protect themselves against floods. It is important to say that even though there were stores that did not experience the three flood events, the area did, so the newest stores learned from those that came before, and started their business with floodgates at least equal in height to those of the previous owner. Therefore, the accumulation of memories resulting from the experiences lived during flood extremes is evident.

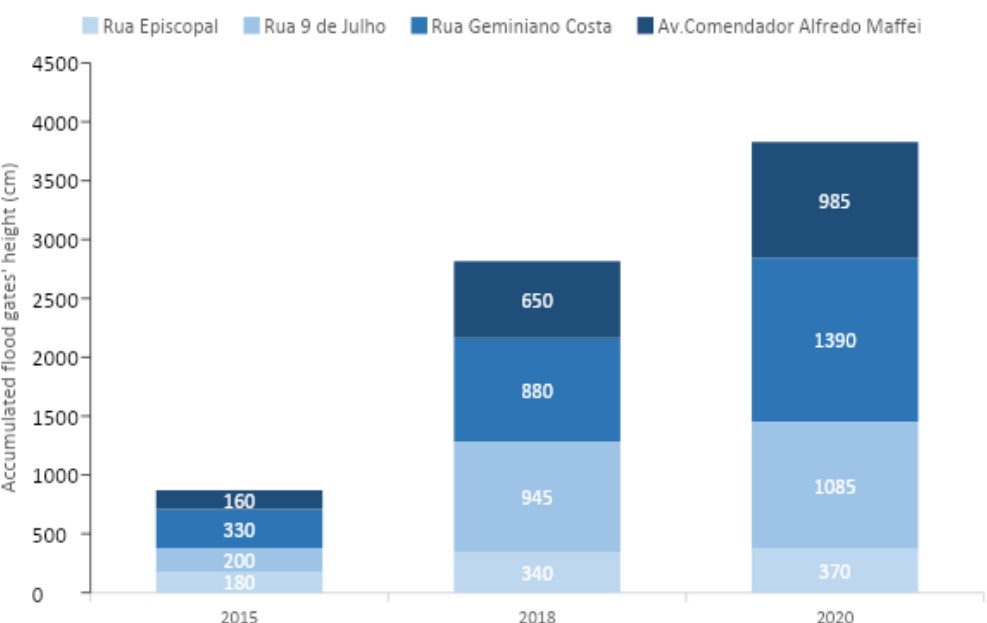

**Figure 8.** Sum of floodgate height separated by the streets considered in this study. The values increased the most in the streets 9 de Julho, Geminiano Costa and Comendador Alfredo Maffei after the event of 2015.

In addition to understanding the recurrence of rainfall, another variable for assessing the magnitude of events is the water level. There are two types of water level to be evaluated: the internal water level, which is the height that the water reaches inside the stores, and external water level, which refers to the water level outside the store or in the street, as verified by the marks of river sediment left on the walls of the establishments. Although the internal and external water level can present the same values, this is not observed in the study area, as most merchants have structures to make it difficult for water to enter their establishment, such as raising the floor in relation to the sidewalk and/or floodgates. Based on this premise, Figure 9 presents the averages of the external and internal water level surveyed in 2019 and 2020, on the basis of a sample of 37, 52 and 23 stores with respect to the 2015, 2018 and 2020 events, respectively, located in the area affected by the floods. We analyzed the heights separately to verify the effectiveness of these structures in the face of flood events. It is important to note that, even though the events of 2015 and 2018 were addressed in the same survey, the number of samples are not the same, because 15 fo the stores had not been in operation in 2015.

The stores within the perimeter of the flood area exhibited a high external water level variation, which was between 83 cm and 170 cm for the 2020 event, 140 cm and 57 cm for the 2018 event, and 110 cm to 60 cm for the 2015 event. Analyzing the internal water level, there was also a difference for the 2020 event; these values oscillated between 31 cm and 138 cm in 2020, between 20 cm and 55 cm in 2018, and between 75 cm and 85 cm in 2015. At the beginning of this section, we highlighted that the events had increasing degrees of magnitude in the following order: 2018, 2015 and 2020. Thus, we would expect water levels to increase in this order. However, this was not the case, due to the important protective role played by floodgates installed in the establishments over the years. The analysis of floodgate heights makes it possible to explain the substantial decrease in the internal water level, as well as to analyze the population's response regarding the new levels of threat faced, especially with regard to the increase in floodgate height as a means of protection following the events experienced by the owners.

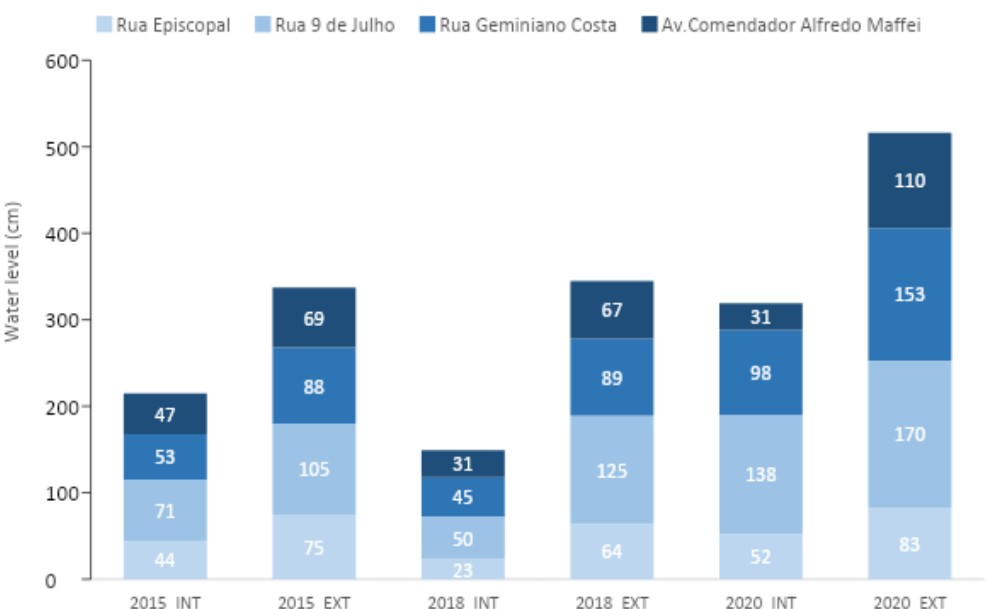

**Figure 9.** Relation between the water levels inside (INT) and outside (EXT) the properties for the three observed flood events, 2015, 2018 and 2020.

The stores within the perimeter of the flood area exhibited a high external water level variation, which was between 83 cm and 170 cm for the 2020 event, 140 cm and 57 cm for the 2018 event, and 110 cm to 60 cm for the 2015 event. Analyzing the internal water level, there was also a difference for the 2020 event; these values oscillated between 31 cm and 138 cm in 2020, between 20 cm and 55 cm in 2018, and between 75 cm and 85 cm in 2015. At the beginning of this section, we highlighted that the events had increasing degrees of magnitude in the following order: 2018, 2015 and 2020. Thus, we would expect water levels to increase in this order. However, this was not the case, due to the important protective role played by floodgates installed in the establishments over the years. The analysis of floodgate heights makes it possible to explain the substantial decrease in the internal water level, as well as to analyze the population's response regarding the new levels of threat faced, especially with regard to the increase in floodgate height as a means of protection following the events experienced by the owners.

By evaluating the heights by street, we can see the differences with respect to the spatial characteristics of the floods. At Rua Nove de Julho, the difference between the average internal and external water level was the most extreme, at around 34 cm for the 2015 event and 75 cm for the event in 2018. It can be inferred that this was due to the preparation against flooding, since, of the 10 establishments visited, 7 had floodgates (70%) in 2015 and after the 2018 event, all stores had installed floodgates or raised them. For 2020, this difference was 32 cm, even with the floodgates; this was due to the magnitude of the event, which reached an external water level of 170 cm. Regarding Rua Comendador Maffei, it is worth noting that, of the nine establishments whose owners were interviewed located on the block between Rua Episcopal and Rua 9 de Julho, none had floodgates in 2015. One of the probable reasons for this is the fact that, according to the merchants, flood events had rarely reached this location in events prior to the year 2015, although these properties are located directly above the Gregório stream, in its buffered portion. Following the 2015 event, floodgates were installed in five establishments in that block, with the others studying its installation, and in 2020, all of the stores whose owners were interviewed had floodgates.

Therefore, it appears that, although the external flood height was greater in the event of 2018 compared to that of 2015, the amount of water that entered the establishments was lower, a fact that can be explained by the greater preparation of the population in relation to the threat of flooding in the form of raising the floodgates. However, the preparation

based on the events of 2015 and 2018 was not enough to prevent water from entering stores in 2020 due to the increased intensity of the event.

### 3.4. Limitations of the Research

The sample size is the main limitation; when the walking interviews were conducted, shopkeepers were not open to talking about the flood, as they were still dealing with their losses. In our methodology, we applied the theory of Ridolfi [7] regarding the four ideal types of society, and to do that we made some assumptions regarding the sample. It is important to note that the use of qualitative data has its own uncertainties (experience, knowledge, loss of memory, changes in policies, and others). We also faced the lack of availability of socio-economic data, since many stores were not willing to share information about this subject; they were still counting their losses, and those that had the information were not sure about the numbers, giving us a range. In acknowledgement of the fact that there are several various factors that may impact the final result, we did not include this within the scope of our study. The authors acknowledge that they may have influenced the responses or the final results on a case-by-case basis.

### 4. Conclusions

The Gregório Creek basin has suffered from flooding for a number of years, and shopkeepers deal with this reality like a risk controlling society, increasing their individual protection as extreme events happen, creating a sense of safety. Only a few of them are aware that there is no way to control nature, and therefore it is necessary to adapt by using soft measures like warning systems, evacuation planning, information sharing, hazard mapping and, in the most critical places, moving to new areas with less risk. On the basis of surveys conducted in 2019 [25] and 2020 addressing three flood events in 2015, 2018 and 2020, it was possible to create a social flood map accurately showing the flood spots of the 2020 event using data volunteered by the affected people.

The results showed that merchants in the study area employ flood protection mechanisms, such as floodgates and having the store level above the sidewalk level; however, due to the magnitude of the event on 12 January 2020, with an accumulation of more than 92 mm precipitation in 3 hours, these measures were not enough in the most critical areas, such as at the corner of Rua Geminiano Costa and 9 de Julho, where the height of the water reached 170 cm above ground level.

The substantial increase in the accumulated height of floodgates and the average height of these gates represents the population's response in the face of new levels of threat. Therefore, this example illustrates the mechanisms of interaction between society and floods, showing the accumulation of memories resulting from lived experience during the extremes of flooding.

Acknowledging the limitation of the sample size, this study highlights the recurrent behavior and the use of anticipated memories for the adaptation of commercial property holders in the basin, as well as representing a practical implementation of theory at the local level and validating it on the ground. A new interview should be conducted to collect more information about the merchants' expectations in the basin in order to better classify the ideal type of society. Future studies should also investigate the efficiency of protection measures and compare them with other practices.

**Author Contributions:** The preparation of this article required the participation of seven researchers, who had their participation as shown below: Conceptualization, H.C.P.F., P.T.S.d.O., E.M.M., B.J.d.O.S., F.G.A. and F.A.A.S.; Methodology, H.C.P.F., P.T.S.d.O., E.M.M., B.J.d.O.S., F.G.A. and N.B.-M.; Formal analysis, H.C.P.F., B.J.d.O.S. and F.A.A.S.; Investigation, H.C.P.F., P.T.S.d.O., E.M.M., B.J.d.O.S., F.G.A. and N.B.-M.; Resourses, H.C.P.F.; Data curation, H.C.P.F., B.J.d.O.S. and F.G.A.; Writing–original draft preparation, H.C.P.F.; writing—review and editing, H.C.P.F., P.T.S.d.O., E.M.M. and N.B.-M.; Project administration, H.C.P.F., P.T.S.d.O., E.M.M., B.J.d.O.S. and N.B.-M.; Funding Acquisition, H.C.P.F., E.M.M. and P.T.S.d.O. All authors have read and agreed to the published version of the manuscript.

**Funding:** This research was supported by grants from the Ministry of Science, Technology, Innovation and Communication (MCTIC) and the National Council for Scientific and Technological Development (CNPq) (Grants 422947/2018-0, 441289/2017-7, and 306830/ 2017-5), and by the Coordenação de Aperfeiçoamento de Pessoal de Nível Superior–Brasil (CAPES) (Finance code 001 and Capes PrInt).

**Institutional Review Board Statement:** Not applicable.

**Informed Consent Statement:** Informed consent was obtained from all subjects involved in the study.

**Data Availability Statement:** Please refer to suggested Data Availability Statements in section "MDPI Research Data Policies" at https://www.mdpi.com/ethics, accessed on 7 February 2020.

**Acknowledgments:** We would like to acknowledge the National Council for Scientific and Technological Development (CNPq) and the Coordenação de Aperfeiçoamento de Pessoal de Nível Superior–Brasil (CAPES) for granting me a scholarship for this research.

**Conflicts of Interest:** The authors declare no conflict of interest.

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
