# Peer review of "Anticipated Memories and Adaptation from Past Flood Events in Gregório Creek Basin, Brazil"

_water, doi:10.3390/w13233394_

Round 1
Reviewer 1 Report
The presented results are although interesting but generally expected. My recommendations are as follows.
- Authors should show more clearly the novelty of their research. Is it conceptual or regional novelty? What is a step forward compared to previous studies (not only in your city)?
- Why did you carry out this study among shopkeepers but not ordinary homeowners? Please explain this issue in the manuscript.
- The manuscript lacks a section devoted to the limitations and uncertainties of the study. It is unlikely that the interpretation of Figure 4 and subsequent ones is so unambiguous. For example, you didn't take into account the annual income of each store. A shopkeeper with a higher income could/can more easily “survive” floods than his poorer colleague. Consequently, it is more profitable for the latter to either close his store or move to another place in the city. These events may well happen regardless of the "historical memory of the floods". There were/are probably few such cases, but they could/can affect your results (including clustering). Please be more critical of your approaches and results.
- What specific recommendations could you give to the studied area of the city based on the results obtained?
Additional comments:
- Figure 2. Write the names of the main streets and indicate the location of the watercourse.
- Line 129. "... on the extension of the floodplain ..."? In the text, you misinterpret the term floodplain. For example, in the caption in Figure 5 - "The dashed line represents the floodplain of the January 12th 2020 flood." River floodplains are relatively stable landforms. What you have highlighted in the figure with a dashed line is a flooded part of the floodplain in the city. Please use hydrological and geomorphological terms correctly! Check it out in the text as well.
- Figure 9 and throughout the text. Water level? In comparison with what was the height of this level calculated? The creek water level during non-flood periods? Explain it in the text (Methods).
Author Response
Reviewer #1: We would like to thank the referee for the insightful comments, suggestions. The manuscript has been revised in accordance with your comments, which enabled us to improve its quality. Below you will find answers to more specific comments.

Reviewer 2 Report
You can use works from the bibliography
Avram E., Armaș I. (2009), Riscul inudațiilor și încrederea socială, Revista de Psihologie aplicată, Editura Universității de Vest din Timișoara, vol.11, nr.2, 89-96.
Baan P J A, Klijn F (2004) Flood risk perception and implications for flood risk management in the Netherlands, Int J River Basin Manag 2:1–10
Bera M K, Daněk P (2018) The perception of risk in the flood-prone area: a case study from the Czech municipality. Disaster Prevention Manag 27(1):2-14
Botzen W J W, Aerts J C J H, Van Den Bergh J C J M (2009a) Dependence of flood risk perceptions on socioeconomic and objective risk factors. Water Resour Res. Doi 10.1029/2009WR007743
Bustillos Ardaya A, Evers M, Ribbe L (2007) What influences disaster risk perception? Intervention measures, flood and landslide risk perception of the population living in flood risk areas in Rio de Janeiro state, Brazil. Int J Disaster Risk Reduct 25:227–237
Comănescu L., Nedelea A., (2016), Floods and public perception on their effect. Case Study: Tecuci Plain (Romania), year 2013, Procedia Environmental Sciences - International Conference – Environment at a Crossroads: SMART approaches for a sustainable future, 32: 190 – 199, doi: 10.1016/j.proenv.2016.03.024
Duží B, Vikhrov D, Kelman I, Stojanov R and Juřička D (2015) Household measures for river flood risk reduction in the Czech Republic. J flood Risk Manag. https://doi.10.1111/jfr3.12132
Działek J, Biernacki W, Bokwa (2013a) Impact of social capital on local communities’response to floods in southern Poland. In Neef A, Shaw R (eds) Risks and Conflicts: Local Responses to Natural Disasters. Community, Environment and Disaster Risk Management Vol. 14, Emereland, Bingley, pp 195-205
Działek J, Biernacki W, Bokwa A (2013b) Challenges to social capacity building inflood-affected areas of southern Poland. Nat Hazards Earth Syst Sci 13:2555–2566
Grothmann T, Reusswig F (2006) People at risk of flooding: Why some residents take precautionary action while others do not. Nat Hazards 38:101–120
Hung H C (2009) The attitude towards flood insurance purchase when respondents' preferences are uncertain: a fuzzy approach. J Risk Res 12(2):239-258
Kellens W, Zaalberg R, Neutens T, Vanneuville W, De Maeyer P (2011) An Analysis of the Public Perception of Flood Risk on the Belgian Coast. Risk Anal 31:1055-1068
Kellens W, Terpstra T, Schelfaut K, De Maeyer P (2013) Perception and
communication of flood risks: A literature review. Risk Anal 33(1):24-49
Knocke E T, Kolivras K N (2007) Flash flood awareness in south west Virginia. Risk Anal 27:155–169
Kreibich H, Thieken A H, Petrow T, Müller M, Merz B (2005) Flood loss reduction of private households due to building precautionary measures: Lessons learned from the Elbe flood in August 2002. Nat Hazards and Earth Syst Sci 5:117–126
Lechowska E., (2018), What determines flood risk perception? A review of factors of flood risk perception and relations between its basic elements, Natural Hazards, DOI: 10.1007/s11069-018-3480-z
Ledoux, B. (2006), La gestion du risqué inondation, Edition TEC& DOC.
Lindell M K, Hwang S N (2008) Household’s perceived personal risk and responses in a multihazard environment. Risk Anal 28:539–556
Ludy J, Kondolf G M (2012) Flood risk perception on lands ‘protected’ by 100-year Levees. Nat Hazards. https://doi.10.1007/s11069-011-0072-6
Miceli R, Sotgiu I, Settanni M (2008) Disaster preparedness and perception of flood risk: A study in an alpine valley in Italy. J Environ Psychol 28:164–173
O’Neill E, Brereton F, Shahumyan H, Clinch J P (2016) The Impact of Perceived Flood Exposure on Flood-Risk Perception: The Role of Distance. Risk Anal 36(11): 2158-2186
Oasim S, Khan A N, Shrestha RP, Qasim M (2015) Risk perception of the people in the flood prone Khyber Pukhthunkhwa province of Pakistan. Int J Disaster Risk Reduct 14:373–378
Pagneux E, G´ıslad´ottir G, J´onsd´ottir S (2011) Public perception of flood hazard and flood risk in Iceland: A case study in a watershed prone to ice-jam floods. Nat Hazards 58:269–287
Poussin J, Botzen W W, Aerts J C (2014) Factors of influence on flood damage mitigation behaviour by households, Environ Sci Policy 40:69–77
Raaijmakers R, Krywkow J, van der Veen A (2008) Flood risk perceptions and spatial multi-criteria analysis: an exploratory research for hazard mitigation. Nat Hazards 46:307-322
Scheuer S, Haase D, Meyer V (2011) Exploring multicriteria flood vulnerability by integrating economic, social and ecological dimensions of flood risk and coping capacity:from a starting point view towards an end point view of vulnerability. Nat Hazards 58(2):731–751
Siegrist M, Gutscher H (2008) Natural hazards and motivation for mitigation behavior: People cannot predict the affect evoked by a severe flood. Risk Anal 28:771–778
Stojanov R, Duží B, Daněk T, Němec D, Procházka D (2015) Adaptation to the Impacts of Climate Extremes in Central Europe: A Case Study in a Rural Area in the Czech Republic. Sustain 7(9):12758-12786
Takao K, Motoyoshi T, Sato T, Fukuzono T (2004) Factors determining residents’preparedness for floods in modern megalopolises: The case of the Tokai flood disaster in Japan. J Risk Res 7:775–787
Terpstra T (2011) Emotions, trust and perceived risk: Affective and cognitive routes to flood preparedness behavior. Risk Anal 31:1658–1675
Thistlethwaite J, Henstra D, Brown C, Scott D (2018) How Flood Experience and Risk Perception Influences Protective Actions and Behaviours among Canadian Homeowners. Environ Manag 61:197 – 208
Thieken A H, Kreibich H, Muller M, Merz B (2007) Coping with floods: Preparedness, response and recovery of flood-affected residents in Germany in 2002. Hydrol Sci J 52 1016–1037
Zaalberg R, Midden C, Meijnders A, McCalley T (2009) Prevention, adaptation, and threat denial: Flooding experiences in the Netherlands. Risk Anal 29: 1759–1778
2. Figure 1- More clear
Author Response
Reviewer #2: We would like to thank the referee for the insightful comments, suggestions. The manuscript has been revised in accordance with your comments, which enabled us to improve its quality. Below you will find answers to more specific comments.

Reviewer 3 Report
The topics are interesting. Some comments are listed below.
- In Fig.2, it's suggested to use different symbols to represent the locations for different events and flood frequency. In short, one for the 2015 event only, one for the 2018 event only, and the other for both events. There is a market around the study area. Why not include the market in the study?
In addition, where is the locations of the 2020 event? Any overlaying of 2015, 2018, and 2020 events?
- From Fig. 3, it can be told that the last questionnaire is about the information of the flood gates of 2015 and 2018 events but not the 2020 events. How do these three events compare?
- In Fig. 4, what does the existing time equal to -1 mean? Log scale? In the y-axis, there are two 1s in the figure. Why?
- Line 250, "being respectively 870 cm, 2.815 cm and 3.830 cm". According to the format of the paper, it should be 870 cm, 2,815 cm and 3,830 cm, apart from the survey of all samples of each event. It's suggested to compute the average or total gate heights for those who experienced the three or two events. Those groups experienced more than one flood and may increase their flood gate height accordingly.
- Fig 8 shows the sum of the flood gates' height separated by the streets for different streets. However, since a small sample is collected for each street, it may not so be representative of the actual value.
- The survey was not conducted immediately for the events of 2015 and 2018. Do they have a good memory of the event of 2015? Maybe it's better to compare the 2018 and 2020 events mainly. And additional information can be added for the trend for the three events.
- Discussion is needed for future researchers. What can be improved for the future and limitations of this study? Are the results of Figure 4 similar to that of reference [7]? Do the two adopts the two variables for analysis?
- What is the main contribution of this study? It's easy to guess that the store will increase the gate's height if the flood becomes severer. If Figure 4 is new, the authors should have more information on this figure
Author Response
Reviewer #3: We would like to thank the referee for the insightful comments, suggestions. The manuscript has been revised in accordance with your comments, which enabled us to improve its quality. Below you will find answers to more specific comments.

Round 2
Reviewer 1 Report
My remaining recommendations.
- Please write the names of the streets and the studied stream on the map in Figure 5.
- Remove the outer frame of Figure 7.
- In the title of the manuscript and throughout the text, write "Gregorio Creek basin" instead of "Gregorio basin". The basin is a broader concept than what refers to rivers, streams, etc.
Author Response
Thank you for your suggestions, I've worked and I finished according to your requests:
- Please write the names of the streets and the studied stream on the map in Figure 5. (Done)
- Remove the outer frame of Figure 7. (Done)
- In the title of the manuscript and throughout the text, write "Gregorio Creek basin" instead of "Gregorio basin". The basin is a broader concept than what refers to rivers, streams, etc. (Done)
Reviewer 3 Report
All comments are addressed.
Figure 2 shows the 52 stores interviewed by Abreu. And the authors interviewed 23 of them later. It is suggested to use different legned for the two.
Author Response
Thank you for your suggestions, I've worked and I finished according to your request:
Figure 2 shows the 52 stores interviewed by Abreu. And the authors interviewed 23 of them later. It is suggested to use different legend for the two. (Done)